

# The closure temperature(s) of zircon Raman dating

Birk Härtel[1], Raymond Jonckheere[1], Bastian Wauschkuhn[1], and Lothar Ratschbacher[1]

[1] Geologie, TU Bergakademie Freiberg, Bernhard-von-Cotta-Straße 2, 09599 Freiberg, Germany

*Correspondence to*: Birk Härtel (haertelb@mailserver.tu-freiberg.de)

**Abstract.** We conducted isochronal and isothermal annealing experiments on radiation-damaged zircons between 500 and 1000 °C for durations between ten minutes and five days. We measured the widths ($\Gamma$) and positions ($\omega$) of the internal $\nu_1(SiO_4)$, $\nu_2(SiO_4)$, $\nu_3(SiO_4)$, and external rotation Raman bands at $\sim$ 974, 438, 1008, and 356 cm$^{-1}$. We fitted a Johnson-Mehl-Avrami-Kolmogorov and a distributed activation energy model to the fractional annealing data, calculated from the widths of the $\nu_2(SiO_4)$, $\nu_3(SiO_4)$, and external rotation bands. From the kinetic models, we

determined closure temperatures $T_c$ for damage accumulation for each Raman band. $T_c$ range from 330 to 370 °C for the internal $\nu_2(SiO_4)$ and $\nu_3(SiO_4)$ bands; the external rotation band is more sensitive to thermal annealing ($T_c \sim$ 260 to 310 °C). Our estimates are in general agreement with previous ones, but more geological evidence is needed to validate the results. The $T_c$ difference for the different Raman bands offers the prospect of a multi-closure-temperature zircon Raman thermochronometer.

## 1 Introduction

Zircon (ZrSiO$_4$) is used with several geochronometers because of the substitution of U and Th for Zr in its lattice. Its occurrence in various types of rocks and high chemical and mechanical resistance make it useful for geochronological applications. The $\alpha$-disintegration of U and Th creates lattice disorder by the impact of $\alpha$-particles and the recoil of daughter nuclei. The zircon Raman spectrum is sensitive to lattice damage: the downshift and broadening of the Raman

bands provide a quantitative measure for the radiation damage (Nasdala et al., 1995, 1998). Measurements of the radiation damage and of the U and Th content allow to calculate a zircon Raman age (Pidgeon, 2014; Jonckheere et al., 2019).

Radiation damage is annealed at high enough temperatures (Zhang et al., 2000a; Geisler et al., 2001; Nasdala et al., 2001; Pidgeon et al., 2016). Figure 1 shows the Raman spectrum of zircon with progressive annealing. The Raman

bands shift to higher wavenumbers, towards the band positions of well-ordered zircon, and become narrower and more intense. This loss of damage with temperature and time is a problem for the interpretation of zircon Raman ages as crystallization ages (Nasdala et al., 2001, 2002) but unlocks the potential for determining cooling ages and analyzing the thermal histories of natural zircon samples (Resentini et al., 2020). Annealing of radiation damage also affects He-diffusion and is thus a process that needs to be taken into account in the interpretation of (U-Th)/He data of zircon

(Ginster et al., 2019; Anderson et al., 2020).

A thermochronometer is characterized by its closure temperature $T_c$, the temperature of the dated sample at the time of its apparent age (Dodson, 1973, 1979). $T_c$ estimates range from ~130 °C for natural samples at isothermal conditions in the KTB borehole (Jonckheere et al., 2019) to ~650 °C for the re-crystallization of metamict zircon based on the retention of Pb in zircons that were heated to these temperatures (Mezger and Krogstad, 1997). Pidgeon (2014) placed

$T_c$ between 230 and 320 °C based on the comparison of Raman ages with other thermochronological data of the same geological units.

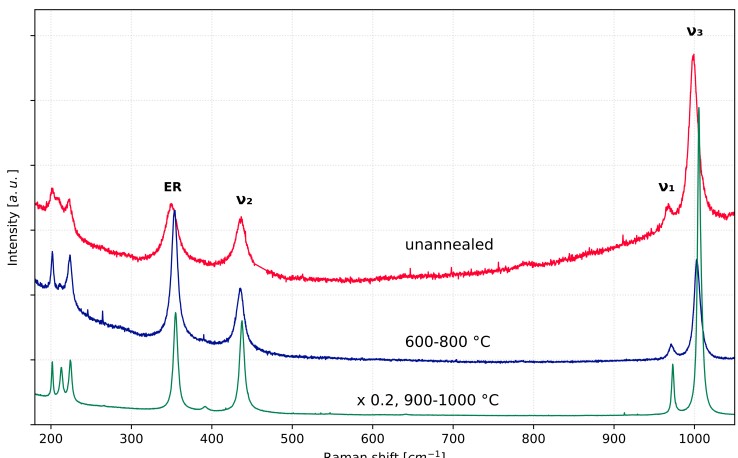

**Figure 1. Raman spectrum of an unannealed, radiation-damaged zircon (red) compared with the same grain after cumulative annealing for 1h at 600, 1h at 700, and 1h at 800 °C (blue), and additional two 1h steps at 900 and 1000 °C (green). The intensities of the latter spectrum are reduced by a factor of 5 for comparison.**

Previous laboratory annealing experiments distinguished several annealing stages based on the changes of lattice constants measured by XRD (Weber, 1993, Colombo and Chrosch, 1998a, 1998b) and on changes in the relationship of Raman shift ($\omega_3$) to bandwidth ($\Gamma_3$) of the $\nu_3(SiO_4)$ Raman band (Geisler et al., 2001, Geisler, 2002, Ginster et al., 2019).

Figure 2 plots $\omega_3$ against $\Gamma_3$ for the experiments of Geisler (2002) and Ginster et al. (2019). The offset and difference in slope between the damage accumulation and annealing trends are evident. Breaks in slope of the annealing trend mark transitions between the annealing stages. A sharp break separates the steep stage I and the flat stage II, but a more gradual transition occurs between stage II and a stage III assumed by Geisler (2002). Stage I is dominated by the elimination of point defects; stage II is ascribed to crystallization of the amorphous domains (Colombo and Chrosch,

1998a, Capitani et al., 2000, Geisler et al., 2001, Geisler, 2002, Ginster et al., 2019); and stage III is related to the diffusion of residual point defects (Geisler, 2002).

Our aim is to investigate the change of the major Raman bands on annealing and to estimate $T_c$. We track the changes of $\omega$ and $\Gamma$ of the $\nu_1(SiO_4)$, $\nu_2(SiO_4)$, $\nu_3(SiO_4)$ internal Raman bands, and the external rotation band at ~ 974, 438, 1008, and 356 cm$^{-1}$ (Kolesov et al., 2001) for isochronal annealing runs at different temperatures. We fit two kinetic models to the

widths of the three most intense Raman bands for isothermal annealing for different time intervals and temperatures and consider their extrapolation to geological timescales. We discuss the closure temperatures calculated from the models in comparison to previous $T_c$ estimates.

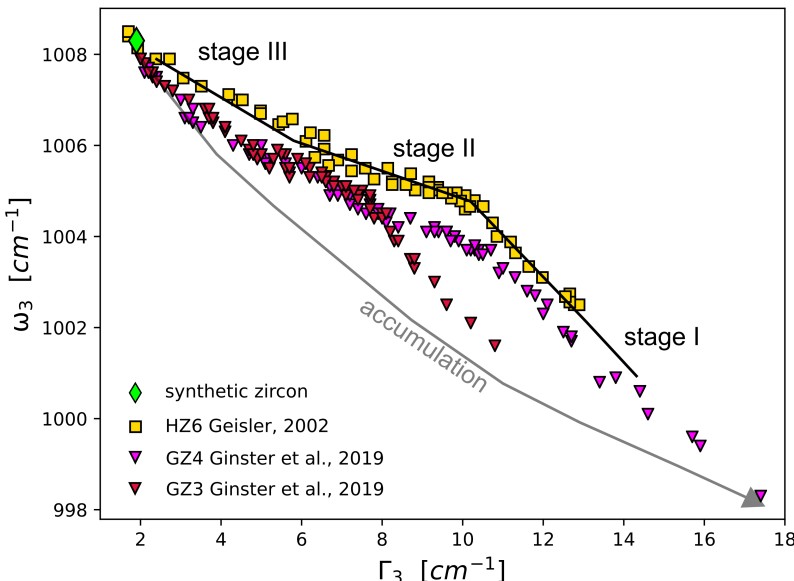

**Figure 2. Position-bandwidth (ω-Γ) plot of the zircon ν₃(SiO₄) band, showing the radiation-damage accumulation trend (grey trajectory) and the stage I - III annealing data of Geisler (2002) and Ginster et al. (2019). The black trajectory shows the difference in slope between the stages leading up to the values of synthetic zircon.**

## 2 Methods and materials

### 2.1 Zircon samples

We separated zircons from a Late Carboniferous volcanic rock from the Flöha Basin in Saxony, Germany (Löcse et al., 2019). This sample was selected because the zircons are assumed to have retained the radiation damage accumulated since their formation. They have moderate to high radiation-damage densities. Zircon separation was carried out as described in Sperner et al. (2014). The zircon grains for the annealing experiments were hand-picked under a binocular

microscope. We discarded grains with cracks that might fall apart upon heating or cooling. For comparison, we measured a zircon synthesized as pure ZrSiO₄ by Guillong et al. (2015).

### 2.2 Raman spectrometry

We measured the Raman spectra using a TriVista Spectrometer (Princeton Instruments) in single mode. The power of the 488 nm incident laser light on the sample is ~12 mW. Repeat measurements show that the laser power does not affect the

lattice damage. The wavenumber calibration used the 219.2, 520.7, and 1001.4 cm⁻¹ bands of sulfur, silicon, and polystyrene. The spectral resolution is ~0.8 cm⁻¹ and the pixel resolution on the detector is ~0.2 cm⁻¹. We acquired zircon spectra in step-and-glue mode with three steps spanning 170 to 1100 cm⁻¹. We cut the spectra into three regions and fitted the bands with Lorentz functions using a 3$^{rd}$ order polynomial for background subtraction. We corrected the bandwidth for the instrumental function following Tanabe and Hiraishi (1980).



**2.3 Annealing experiments**

We performed isothermal and isochronal annealing runs in a Linn LM111.06 and a Nabertherm LT3/11 muffle oven. For each run, we prepared a set of zircon grains showing different radiation damage to be annealed together. The zircon grains were individually wrapped in Monel 400 foil (a nickel/copper alloy) and inserted in the pre-heated oven in a ceramic crucible. The temperatures of the isothermal annealing runs ranged from 500 to 1000 °C for cumulative

annealing times of 10 minutes, 30 minutes, 5 hours, 24 hours, and 5 days. The experiments followed the approach of Geisler et al. (2001) with each zircon being annealed in consecutive steps at a constant temperature and cooled for measurement between the steps.

In the isochronal experiments, we annealed the zircons for one hour runs at 600 to 1000 °C with a 100 °C interval. The Raman spectrum was measured at room temperature after each run. The locations of measurement spots on the zircon

grains were recorded before each annealing step to assure measurements at the same locations. Grains that disintegrated during annealing were discarded.

**3 Results and Discussion**

**3.1 Changes in band position and width**

The change in band position, bandwidth, and intensity is different for each Raman band (Figure 1), as reported in earlier

studies (Zhang et al., 2000a; Geisler, 2002). Figure 3 shows plots of ω *vs.* Γ for the isochronal annealing runs.

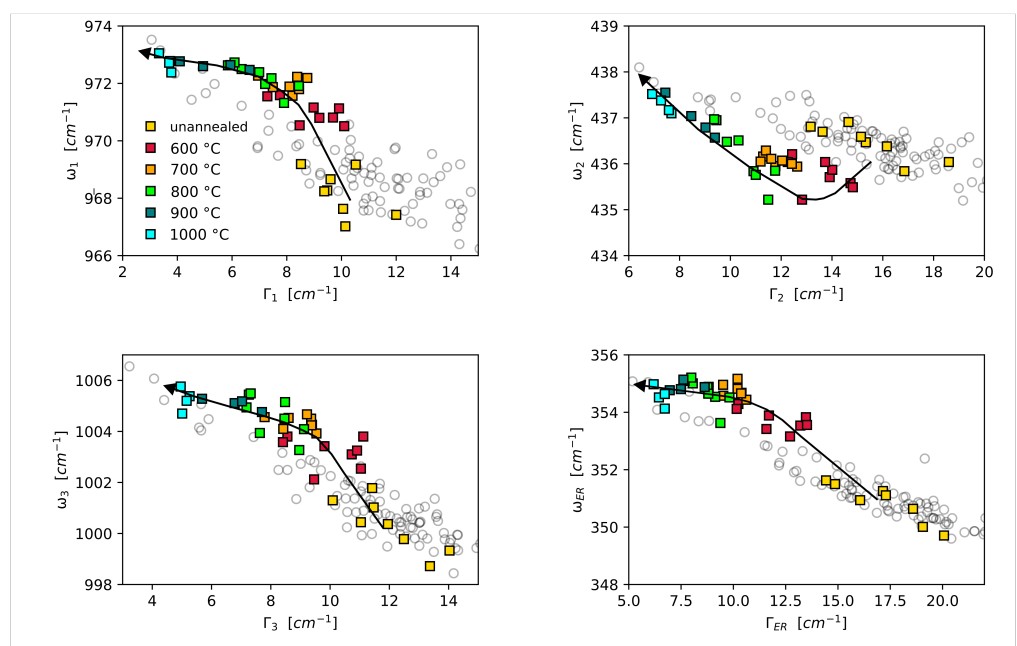

**Figure 3. ω-Γ plots of the ν₁(SiO₄) ν₂(SiO₄) ν₃(SiO₄) and external rotation (ER) Raman bands for the isochronal annealing runs. Gray circles represent the unannealed zircons from which the annealed samples were selected.**





The different bands exhibit a common trend of decreasing Γ and increasing ω with increasing temperature, but trace distinct lines through the ω-Γ space. Overall, the ω-Γ trajectories for the $v_1(SiO_4)$, $v_3(SiO_4)$, and external rotation bands resemble that of $v_3(SiO_4)$ in Figure 2 with a steep segment at the beginning, followed by a break in slope towards a

flatter trend. $\omega_3$ and $\Gamma_{ER}$ show the largest changes. The breaks in slope between 700 and 800 °C for the 1h annealing runs are interpreted as the transition from stage I to stage II annealing (Geisler et al., 2001). The slopes of the stage I and stage II segments are different for each band. A striking difference exists between the $v_2(SiO_4)$ band near 438 cm$^{-1}$ and the other bands. During stage I, $\omega_2$ decreases, whereas the other three bands shift to higher wavenumbers. The decrease of $\omega_2$ reverses at higher temperatures at the onset of stage II of the other bands. $\Gamma_2$ values decrease throughout

the annealing process like the other bandwidths. The scatter of the ω-Γ data around each common trend is limited, producing a well-defined trend for all Raman bands for stage II, irrespective of the different radiation-damage densities in the unannealed zircons. Our data do not show the gradual steepening between the stages II and III as in Figure 2.

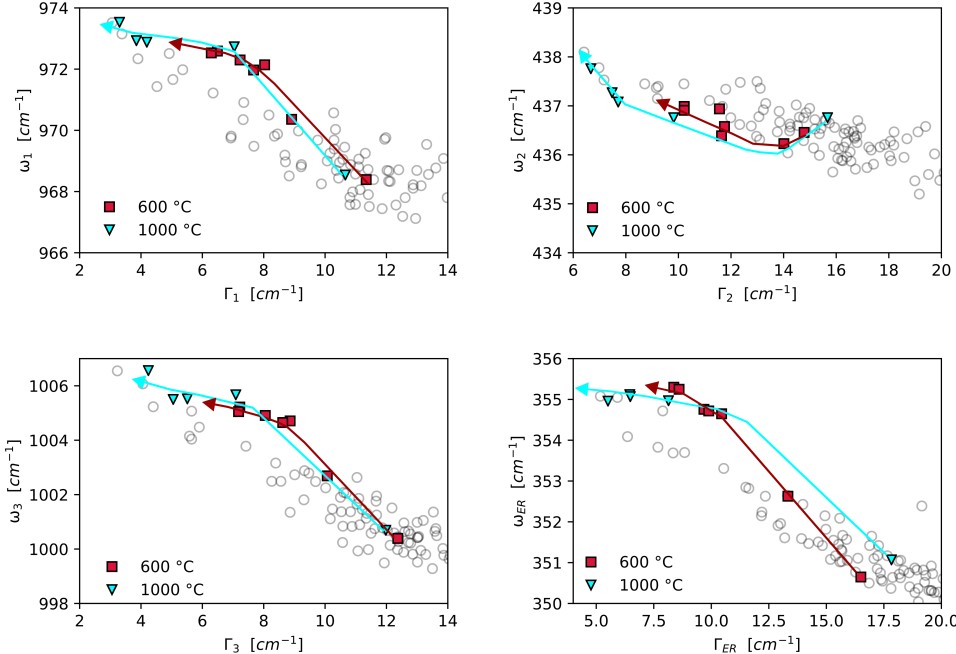

**Figure 4. ω-Γ plots of the $v_1(SiO_4)$ $v_2(SiO_4)$ $v_3(SiO_4)$ and external rotation (ER) Raman bands for the isothermal annealing runs at 600 and 1000 °C. Gray circles represent the unannealed zircons from which the annealed samples were selected.**

Figure 4 traces the ω-Γ trends for two zircon grains with similar initial radiation damage through isothermal annealing at 600 and 1000 °C. As expected, annealing proceeds faster at 1000 °C. The ω-Γ trends follow the same trajectories as

in Figure 3. The stage I sections of the 1000 °C trajectories must be assumed because the first annealing step already reached stage II.

Figure 5 compares the change in band positions of the isochronal runs with the results of Zhang et al. (2000a), and in the $\omega_3$ position of Geisler (2002) and Ginster et al. (2019; Figure 5c). The $\omega_1$, $\omega_3$, and $\omega_{ER}$ data define a rising trajectory

up to 800 °C and a flat annealing trend at higher temperatures. This reflects the trends for stage I and II (Figures 2, 3, 4).

For $\omega_3$, most of our data are slightly lower than those of Ginster et al. (2019), consistent with the values of Geisler (2002), but more strongly annealed than those of Zhang et al. (2000a). We also observed more annealing of $\omega$ compared with Zhang et al. (2000a) for the other three bands. The small difference between our isochronal runs for 1h and the 90 min runs of Ginster et al. (2019) can be attributed to the difference in annealing time.

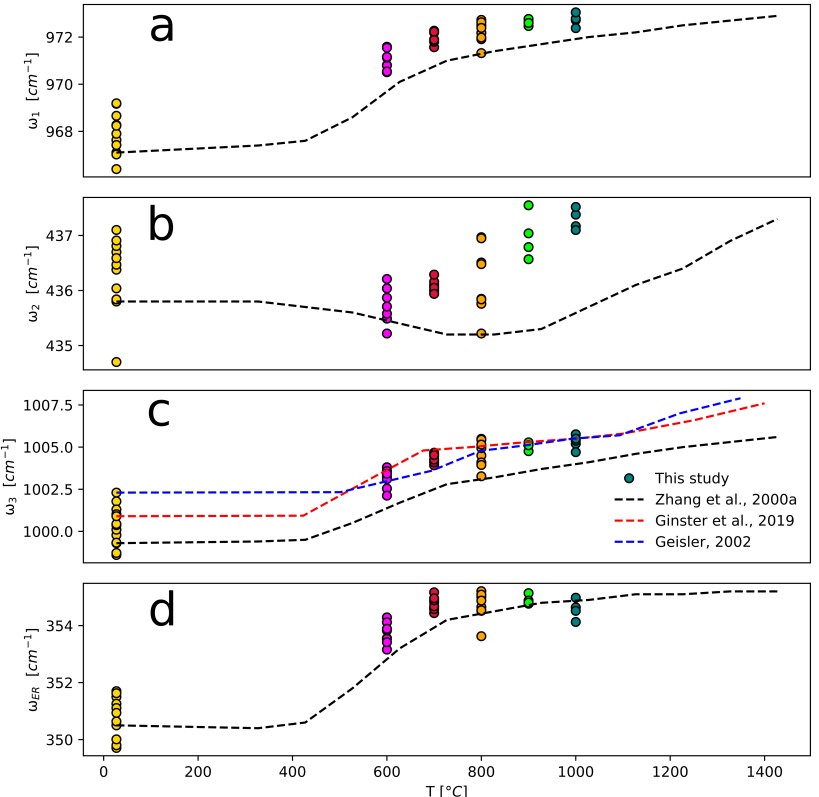

**Figure 5. $\omega$-temperature plots of the $\nu_1(SiO_4)$, $\nu_2(SiO_4)$, $\nu_3(SiO_4)$, and external rotation (ER) Raman bands for the isochronal experiments compared with the 1h isochronal annealing data of Zhang et al. (2000a), and the 90 min annealing experiments of Geisler (2002) and Ginster et al. (2019).**

The main difference between our results and those of Zhang et al. (2000a), Geisler (2000), and Ginster et al. (2019)
relates to $\omega_2$, for which Geisler (2002) reported a constant value through stage I and II that only begins to increase in stage III. Our data show an initial drop of $\omega_2$, followed by a shift to higher wavenumbers at the onset of stage II. Zhang et al. (2000a) show a similar decrease of $\omega_2$ which was only reversed during stage III.

The differences between $\nu_2(SiO_4)$ and the other Raman bands are interpreted as due to the different Raman modes. The vibrational frequencies of the stretching bands $\nu_1(SiO_4)$ and $\nu_3(SiO_4)$ depend most on the bond lengths. In contrast,
$\nu_2(SiO_4)$ is a bending mode, depending on the angle between the Si-O bonds in the $SiO_4$ tetrahedron (Geisler, 2002). The O-Si-O angle is related to the ratio of the unit cell parameters $a$ and $c$ (Tokuda et al., 2019). Figure 6 plots $c$ *vs. a* for the XRD data for the annealing experiments of Colombo and Chrosch (1998a) and Geisler et al. (2001). Due to damage



accumulation, *c* increases more than *a* (Salje et al., 1999; Zhang et al., 2000b). The increased ratio *c/a* reduces the O-Si-O angle between the oxygen atoms shared by Si and Zr (Tokuda et al., 2019), shifting the $\nu_2(SiO_4)$ Raman band to lower
wavenumbers. During stage I annealing, the unit cell shrinks anisotropically, reducing *a* more than *c*, causing *a* further increase of *c/a* and lowering of the O-Si-O angle. The anisotropic shrinkage is thought to be due to the preferential diffusion of point defects in the basal plane of zircon during recovery (Ríos et al., 2000; Colombo and Chrosch, 1998a). We interpret the further decrease of the O-Si-O angle to cause of the decrease of $\omega_2$ during stage I. The annealing trend in the *c vs. a* plot changes to a preferential reduction of *c* during stage II until the values of well-ordered zircon are
reached. The decrease of the *c/a* values is accompanied by the opening of the O-Si-O angle, which we associate with the reversal of $\omega_2$ during stage II.

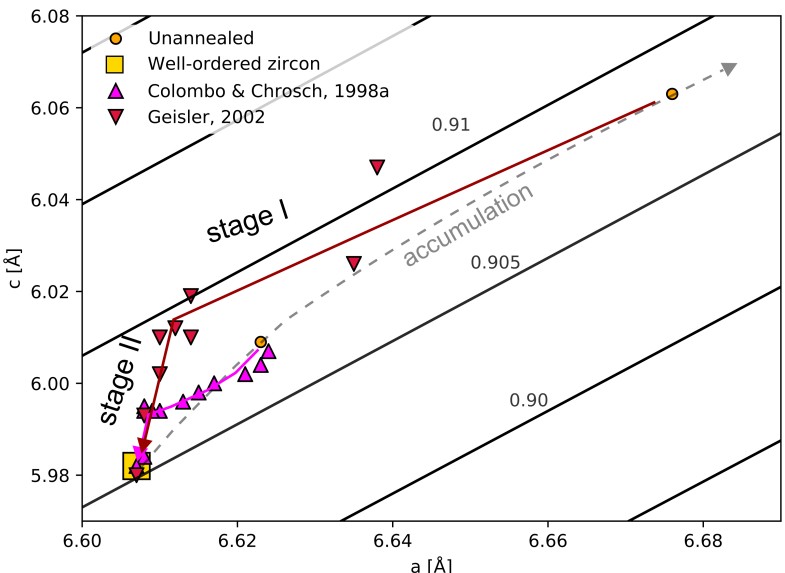

**Figure 6. Zircon unit cell measurements of Colombo and Chrosch (1998a) and Geisler (2002). The dashed line represents the radiation-damage accumulation trend and the colored lines the annealing trajectory, starting from different damage densities. The black lines indicate constant *c/a* ratios.**

Figure 7 plots $\omega_3$ *vs.* $\Gamma_3$ for our isochronal and isothermal runs, superimposed on the data of Geisler et al. (2001), Geisler (2002), and Ginster et al. (2019). From stage II on, the annealing data define a well-defined common trend, even for zircon samples with different initial damage densities. The samples trace sub-parallel trajectories through stage I. The
convergence towards a common stage II is also apparent for the other Raman bands (Figures 3 and 4), as well as for the XRD unit cell data (Figure 6). Stage II is interpreted as representing a state of the zircon lattice that is independent of the damage accumulation history. We assume, based on the interpretation of the successive annealing stages of Colombo and Chrosch (1998a), Ríos et al. (2000), and Geisler et al. (2001), that stage II describes zircons in which the lattice has lost most of its point defects and is predominantly strained by the amorphous domains caused by α-recoils. In
this case, the position of a zircon along the stage II trend represents the remaining amorphous fraction.

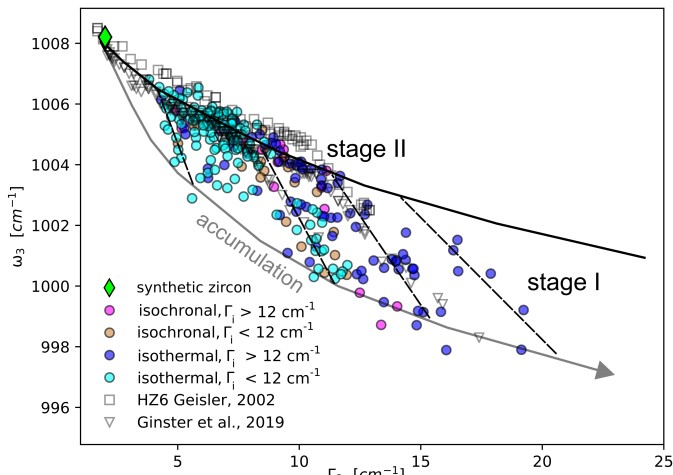

**Figure 7. Composite ω-Γ plot of the ν₃(SiO₄) annealing data from this study compared with published data. The data are subdivided into isochronal and isothermal runs and samples with initial $\Gamma_3 \lesssim 12$ and $\gtrsim 12$ cm⁻¹. Gray line: Radiation-damage accumulation trajectory, dashed lines: stage I annealing, bold line: stage II annealing.**

## 3.2 Kinetic modeling and closure temperature

For estimating the temperatures at which annealing takes place on geological timescales, we fitted kinetic models to the Raman bandwidth data for the isothermal annealing runs. We fitted $\Gamma_2$, $\Gamma_3$, and $\Gamma_{ER}$, but not $\Gamma_1$ which shows lower

bandwidths than the other bands, implying lower sensitivity to radiation damage. We quantified the fractional lattice repair $\Phi(t, T)$ following isothermal annealing for a time $t$ and a temperature $T$ equivalent to the parameter α of Geisler et al. (2001):

$$\Phi\left(t, T\right) = \frac{\Gamma_i - \Gamma\left(t, T\right)}{\Gamma_i - \Gamma_0} \tag{1}.$$

$\Gamma_i$ is the bandwidth of the unannealed sample, $\Gamma(t,T)$ that after annealing for a (cumulative) time $t$ at temperature $T$. $\Gamma_0$ is

the bandwidth of undamaged zircon; we assumed 5.0, 1.9, and 3.6 cm⁻¹ for the $\nu_2(SiO_4)$, $\nu_3(SiO_4)$, and external rotation bands, based on the values of the synthetic zircon, we measured. $\Phi = 0$ indicates no annealing, $\Phi = 1$ complete annealing. A Pearson correlation test with Bonferroni correction for multiple testing (Abdi, 2007) showed that none of the co-annealed zircons exhibited a significant dependence of $\Phi$ on the initial damage $\Gamma_i$. We fitted the arithmetic mean $\Phi(t, T)$ values of each experimental condition $(t, T)$ to approximate equal weighting of the different isothermal

annealing runs.



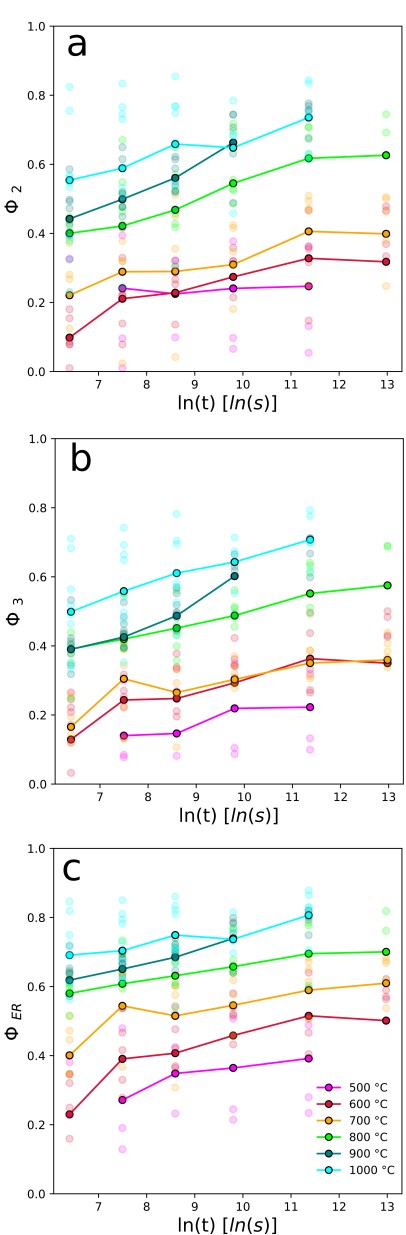

**Figure 8. Plots of the annealed fraction Φ against ln annealing time for the $\nu_2(SiO_4)$ (a), $\nu_3(SiO_4)$ (b), and external rotation (c) Raman bands. The arithmetic means are connected with lines for visual guidance.**

Figure 8 plots $\Phi$ against annealing time for the three Raman bandwidths; $\Phi$ ranges from ~0.1 to ~0.7 for $\Gamma_2$ and $\Gamma_3$ and from ~0.2 to ~0.8 for $\Gamma_{ER}$. The trends are approximately linear with logarithmic time and roughly parallel to each other. As expected, $\Phi$ increases with time and temperature. The values for $\nu_3(SiO_4)$ are consistent with those of Ginster et al.





(2019), who worked with samples of different age and provenance so that we assume that the annealing kinetics of our

samples are applicable to a broader range of zircons. We fitted two models: a Johnson-Mehl-Avrami-Kolmogorov (JMAK) model (Kolmogorov, 1937; Avrami, 1939; Johnson and Mehl, 1939) and a distributed activation energy (DAE) model (Lakshmanan et al., 1991; Lakshmanan and White, 1994). The JMAK model is described by:

$$\Phi\left(t\,,\,T\right) = 1 - \exp\left[-\left(kt\right)^n\right] \tag{2},$$

where $n$ is the Avrami exponent and $k$ is a temperature-dependent rate factor that follows an Arrhenius law:

$$k = k_0\,\exp\left(\frac{-E_A}{\kappa T}\right) \tag{3}.$$

$k_0$ is a frequency factor, $E_A$ an activation energy, and $\kappa$ the Boltzmann constant. JMAK models are used for describing crystallization processes (Avrami, 1939; Johnson and Mehl, 1939). Since crystallization of amorphous domains takes place during radiation-damage annealing, Geisler et al. (2001) and Geisler (2002) used this model for estimating the activation energies of the radiation-damage annealing stages II and III of zircon. The DAE model assumes that the

annealing process draws from a distribution of activation energies. It is applied to processes involving sub-reactions with different activation energies and has been used for describing hydrocarbon decomposition and fission-track annealing (Lakshmanan et al., 1991; Lakshmanan and White, 1994). The fractional repair is expressed as follows:

$$\Phi\left(t\,,\,T\right) = 1 - \int_0^\infty G\left(E\right)\exp\left[-t\,k_0\,\exp\left(\frac{-E}{\kappa T}\right)\right]dE \tag{4}.$$

$k_0$ is a frequency factor and $G(E)$ a Gaussian distribution of activation energies $E$ with mean $E_0$ and standard deviation $\sigma$.

We fitted both models by minimizing the sum squared $\Phi$-residuals (SSR). Table 1 lists and Figure 9 shows the results. The (mean) activation energies are between 2.7 and 3.0 eV for the three bands and both models. In contrast, the $k_0$ values span three orders of magnitude. The Avrami exponent is similar for $\Gamma_2$ and $\Gamma_3$ ($n = 0.11$) and lower for $\Gamma_{ER}$ ($n = 0.08$). The standard deviations of $G(E)$ are ~1 eV for the three Raman bands. The best-fit SSR are comparable for all models with the lowest values for $\Gamma_{ER}$. The overall agreement of predicted and measured $\Phi$ values is close to 1:1 for

all experimental conditions (Figure 9). The SSR surfaces plotted against log $k_0$ and mean $E_A$ show a distinct trough of low SSR that includes the best-fit parameters.

**Table 1. Parameter estimates for the Johnson-Mehl-Avrami-Kolmogorov (JMAK) and distributed activation energy (DAE) annealing models and calculated closure temperatures $T_c$ for cooling rates of 10 and 30 K/Myr.**

| Raman parameter | Model | $E_{A/0}$ [eV] | lg $k_0$ [lg s$^{-1}$] | $n$ | $\sigma$ [eV] | SSR | $T_c$ [°C] at 10 [K Myr$^{-1}$] | $T_c$ [°C] at 30 [K Myr$^{-1}$] |
|---|---|---|---|---|---|---|---|---|
| $\Gamma_2$ | JMAK | 2.9 | 7.8 | 0.11 | - | 0.074 | 368 | 381 |
| $\Gamma_3$ | JMAK | 2.7 | 6.5 | 0.11 | - | 0.059 | 359 | 373 |
| $\Gamma_{ER}$ | JMAK | 2.9 | 9.56 | 0.08 | - | 0.038 | 312 | 323 |
| $\Gamma_2$ | DAE | 3.0 | 9,4 | - | 1.0 | 0.064 | 333 | 344 |
| $\Gamma_3$ | DAE | 2.9 | 8.5 | - | 1.0 | 0.052 | 334 | 346 |
| $\Gamma_{ER}$ | DAE | 2.7 | 9.9 | - | 1.2 | 0.040 | 263 | 273 |





We estimate the closure temperatures $T_c$ with the approach of Dodson (1979) for fission-tracks:

$$t_{50} = \frac{-\kappa T_c{}^2}{E\,(dT/dt)} \tag{5}.$$

The equation considers cooling through the closure temperature following a linear increase of $1/T$ with time; $t_{50}$ is the time at which half the damage is retained. $E$ is the activation energy ($E_A$; JMAK) or its distribution ($G(E)$; DAE). Equation (5) is equated to the model Eqs. (2), (3), and (4), rearranging and substituting 0.5 for $\Phi(t, T)$ (Appendix A):

$$\frac{E_A}{\kappa\,T_c} = \ln\left[\frac{-\kappa\,T_c{}^2\,k_0}{E_A\,(dT/dt)\,\sqrt[n]{\ln 2}}\right] \tag{6}$$

for the JMAK model and

$$\int_0^\infty G\,(E)\,\exp\left[\frac{\kappa T_c{}^2\,k_0}{E\,(dT/dt)}\,\exp\left(\frac{-E}{\kappa T_c}\right)\right]dE = 0.5 \tag{7}$$

for the DAE model.

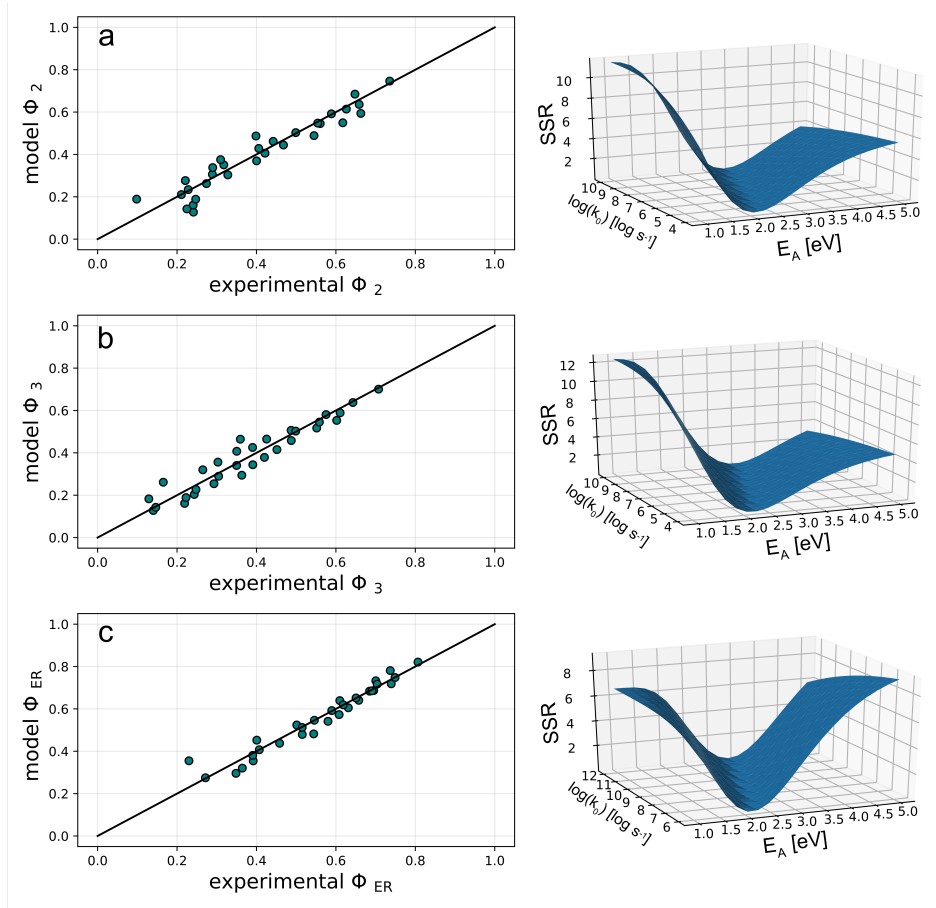



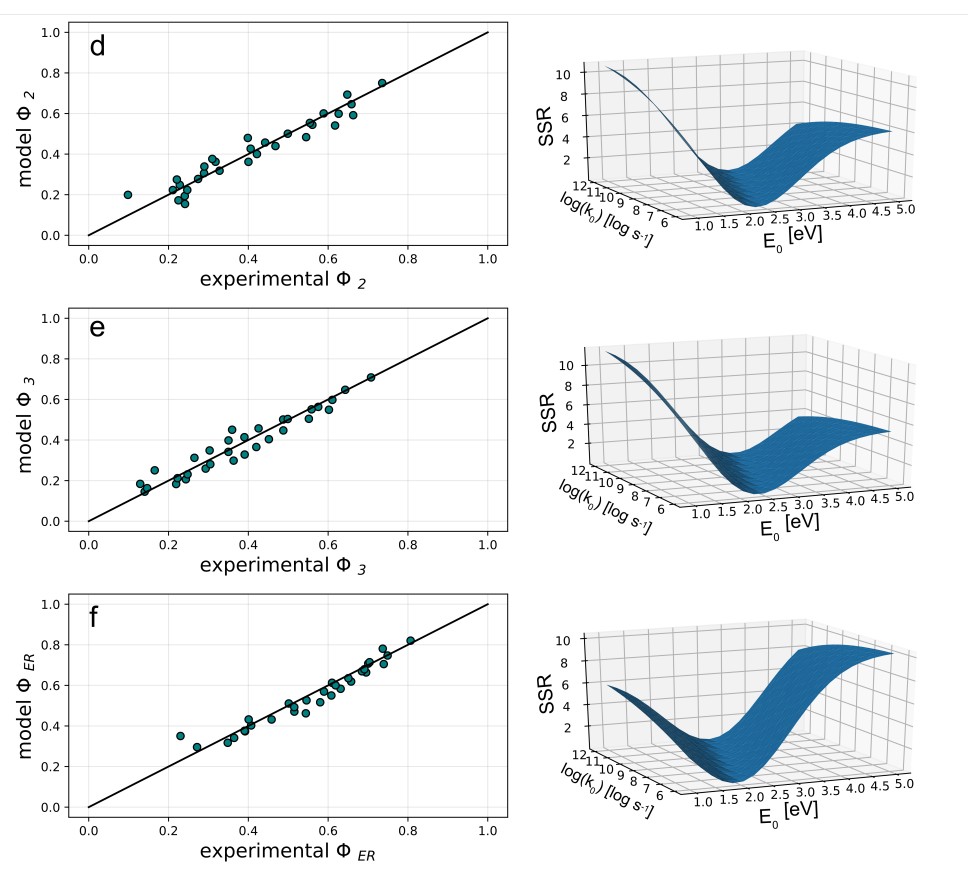

**Figure 9. Overview of the Johnson-Mehl-Avrami-Kolmogorov (JMAK) (a-c) and distributed activation energy (DAE) (d-f) modeling results for $\Gamma_2$, $\Gamma_3$, and $\Gamma_{ER}$. The left panels compare the predicted and measured $\Phi$ values. The right panels show the sum of squared residuals (SSR) surfaces as a function of $\log k_0$ and $E_A$ (JMAK) or $E_0$ (DAE) for optimal values of $n$ and $\sigma$ (Table 1).**

Equations (6) and (7) are solved iteratively for $T_c$. The results for cooling rates of 10 K/Myr and 30 K/Myr are given in Table 1. As expected, $T_c$ is higher for the faster cooling; the difference is ~12 °C. Values for $T_c$ at 10 K/Myr cooling rate range from 260 to 370 °C. The $T_c$ values for all Raman bands are higher for the JMAK than for the DAE models. For both, $T_c$ is highest for $\Gamma_2$ and lowest for $\Gamma_{ER}$; $T_c$ for $\Gamma_3$ is slightly lower than for $\Gamma_2$. We interpret the more sensitive response of the external rotation band to annealing compared with $\nu_2(SiO_4)$ and $\nu_3(SiO_4)$ to the stronger Si-O bonds

within the $SiO_4$ tetrahedra and weaker Zr-O bonds between the tetrahedra (Dawson et al., 1971).

     The linear troughs in Figure 9 reflect a trade-off between $E_A$ ($E_0$ for DAE models) and $k_0$. Different parameter pairs fit the data equally well due to the limited range of laboratory annealing times and temperatures (Mialhe et al., 1988; Lakshmanan et al., 1991). The trade-off is a problem for the extrapolation of the experimental data to geological timescales, since $T_c$ varies along the trough.





The kinetic parameters of our JMAK model for $\Gamma_3$ can be compared to the results of the Geisler et al. (2001) and Geisler (2002) JMAK models for stage II and III annealing. Their Avrami exponent ($n = 0.11$) agrees with ours for $\Gamma_3$. Their activation energy is ~3.8 eV for stage II and ranges from 6.4 to 6.9 eV in stage III; their log $k_0$ is 9.3 for stage II and >15 in stage III. The differences result mainly from the trade-off between $E_A$ ($E_0$) and log $k_0$ and do not necessarily reflect different kinetics. Moreover, most of our data are from annealing stages I and II, whereas those of Geisler et al. (2001)

and Geisler (2002) are from stage II and stage III. Their results and those of Ginster et al. (2019) suggest that stage I annealing requires a lower activation energy than stages II and III which could also in part account for the lower activation energies obtained from our models. The variation of $T_c$ along the SSR troughs in Figure 9 is also the probable reason for the different $T_c$ estimates for the JMAK and DAE models. There is no independent physical evidence for either model, and both models fit our experimental data equally well (Table 1). Therefore, we assume the DAE value as

the lower limit and the JMAK value as the upper limit of the $T_c$ range for each Raman band.

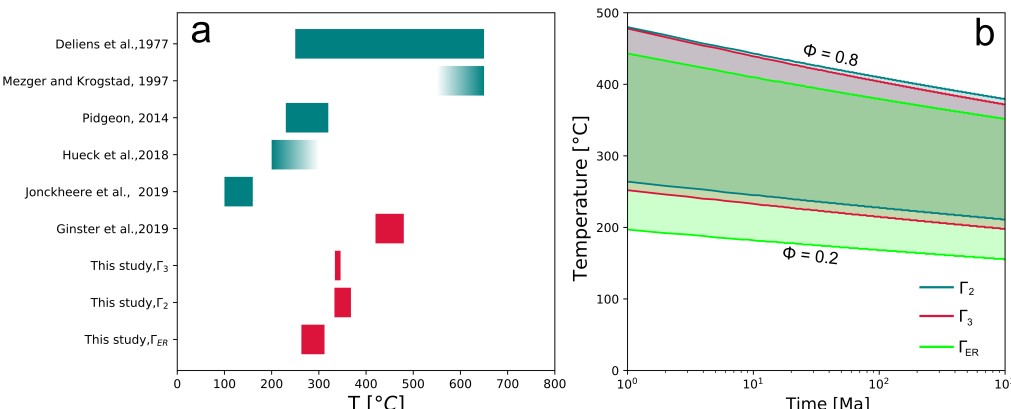

**Figure 10. a) Zircon radiation-damage closure temperatures $T_c$ based on annealing experiments (red) and geological data (green). b) Partial annealing zones for the JMAK models for $\Gamma_2$, $\Gamma_3$, and $\Gamma_{ER}$, for residence times between 1 Ma and 1 Ga.**

Figure 10a compares our $T_c$ values with previous estimates from geological and experimental evidence. The wide range of $T_c$ (160 to 650 °C) is in part due to the different approaches. That of Deliens et al. (1977) resulted from comparing radiation-damage ages of Precambrian zircons, calculated from an internal bending IR band, with the ages determined with established geochronometers. The IR ages tended to be higher than the corresponding titanite U-Pb ages

($T_c \gtrsim 650$ °C, Stearns et al., 2015) and whole-rock Rb-Sr ages, but were mostly higher than mica and feldspar Rb-Sr ages ($T_c \sim 320$ to 575 °C, Harrison and McDougall, 1980; Giletti, 1991). The zircon radiation-damage $T_c$ estimate of Mezger and Krogstad (1997) is based on their observation that zircons that remained below 600 to 650 °C during parts of their geological history experienced Pb-loss by Pb-leaching from metamict zones.

Jonckheere et al. (2019) measured $\Gamma_3$ for isothermal holding for ~80 Myr at increasing temperatures in the KTB

borehole and interpreted its downhole decrease as due to stage I annealing. Hueck et al. (2018) and Pidgeon (2014) dated zircons with Raman based on the $\Gamma_3$ vs. radiation-dose calibration of Palenik et al. (2003). Hueck et al. (2018) compared their results with corresponding (U-Th)/He ages ($T_c \approx 170$ to 190 °C, Reiners et al., 2004) and age-eU modeling results, finding that their Raman ages were consistently higher than the (U-Th)/He ages. Pidgeon (2014) dated zircons from various Australian Precambrian rocks, whose Raman dates were consistent with the biotite Rb-Sr cooling





ages ($T_c \sim 320$ °C, Harrison and McDougall, 1980) for the same units. Pidgeon (2014) placed the onset of stage I zircon radiation-damage annealing at ~230 °C. We calculated a closure temperature for the three $\Gamma_3$ fanning-linear Arrhenius models of Ginster et al. (2019) at 50% damage retention. This gives $T_c$ values between 420 and 480 °C. Our model estimates range from 330 to 370 °C for $\Gamma_2$ and $\Gamma_3$, reflecting the model-dependent extrapolation of the experimental data to geological timescales.

The lower $T_c$ (260 to 310 °C) for $\Gamma_{ER}$ suggest that geological zircon radiation-damage annealing cannot be described by a single $T_c$. Instead, different Raman bands record different parts of the thermal history of a zircon. The dearth of independent experimental and geological data for $\Gamma_2$ and $\Gamma_{ER}$ makes it difficult to be certain that their closure temperatures are different from that of $\Gamma_3$, as the annealing data suggest. For the best studied Raman parameter $\Gamma_3$, our experimental data favor a closure temperature between 330 and 360 °C, in the region between the estimates of Pidgeon

(2014) and Deliens et al. (1977).

Figure 10b shows the partial annealing zones for the JMAK models for $\Gamma_2$, $\Gamma_3$, and $\Gamma_{ER}$. The partial annealing zone temperatures are highest for $\Gamma_2$ and lowest for $\Gamma_{ER}$. Under isothermal holding for >1 Ma, partial annealing occurs at temperatures as low as 200 °C, and full annealing requires temperatures above 450 °C. The low-temperature boundary is in agreement with the stage I annealing temperature of Pidgeon (2014) but higher than that of Jonckheere et al. (2019); the

upper boundary is consistent with full annealing at 600-650 °C assumed by Mezger and Krogstad (1997).

## 4 Conclusions

The results of our isochronal and isothermal annealing experiments indicate that the $\nu_1(SiO_4)$, $\nu_2(SiO_4)$, $\nu_3(SiO_4)$, and external rotation Raman bands at 974, 438, 1008, and 356 cm$^{-1}$ of radiation-damaged zircon anneal differently with respect to the bandwidth ($\Gamma$) and band position ($\omega$). $\Gamma$ decreases for all Raman bands during annealing while $\omega$

increases, but $\omega_2$ drops to lower wavenumbers during the first annealing stage, increasing again from the second annealing stage onwards. The different annealing trajectories can help to detect partial annealing in natural zircons.

Our $\nu_3$ annealing data on volcanic zircons are consistent with those of Ginster et al. (2019), obtained on samples with different provenance, age, and thermal history. This suggests that the results of the annealing experiments are representative for a wide range of natural zircons.

The Johnson-Mehl-Avrami-Kolmogorov and distributed activation energy models yield closure temperatures between 260 and 370 °C. Overall, this range overlaps with most earlier estimates. The different $T_c$ values for both models show that model selection and the trade-offs between the model parameters play a significant role for the extrapolation of laboratory annealing data to geological timescales. This uncertainty in extrapolation emphasizes the need for geological data to constrain $T_c$. Independent of the model, the calculated $T_c$ is comparable for $\Gamma_2$ and $\Gamma_3$ (330 to 370 °C) but lower

for $\Gamma_{ER}$ (260 to 310 °C). This difference offers the prospect of multi-$T_c$ zircon Raman dating using several Raman bands.



## Appendix A: Closure temperature equations

Dodson (1973) defined the closure temperature $T_c$ as the temperature of a parent-daughter system at the apparent age of a rock. For fission tracks, Dodson (1979) equates the time for 50 % annealing at the closure temperature to the cooling time constant defined for a cooling following a linear increase of 1/T:


$$t_{50} = \frac{- \kappa T_c{}^2}{E_A \left( dT / dt \right)}$$

(A1).

$\kappa$ is the Boltzmann constant and $E_A$ is the activation energy of the annealing process at 50% annealing. For the Johnson-Mehl-Avrami-Kolmogorov model, the fraction of annealing $\Phi(t, T)$ is given by:

$$\Phi \left( t , T \right) = 1 - \exp \left[ - \left( t \, k_0 \, \exp \left( \frac{- E_A}{\kappa T} \right) \right)^n \right]$$

(A2).

$k_0$ is a frequency factor, $E_A$ the activation energy and $n$ the Avrami exponent.

Rearranging (A2):

$$- \left( t \, k_0 \, \exp \left( \frac{- E_A}{\kappa T} \right) \right)^n = \ln \left[ 1 - \Phi \left( t , T \right) \right]$$

(A3),

$$\frac{E_A}{\kappa T} = \ln \left[ \frac{t \, k_0}{\sqrt[n]{\ln \left( \frac{1}{1 - \Phi \left( t , T \right)} \right)}} \right]$$

(A4).

Substituting (A1) for $t$ and 0.5 for $\Phi(t, T)$ yields:

$$\frac{E_A}{\kappa T_c} = \ln \left[ \frac{- \kappa T_c{}^2 k_0}{E_A \left( dT / dt \right) \sqrt[n]{\ln 2}} \right]$$

(A5).

(A5) is solved iteratively for $T_c$, given the model parameters $E_A$, $k_0$, and $n$ in Table 1, and assuming a cooling rate $dT/dt$.

For the distributed activation energy model, $\Phi(t, T)$ is given by:

$$\Phi \left( t , T \right) = 1 - \int_0^\infty G \left( E_0 , \sigma \right) \exp \left[ - t \, k_0 \, \exp \left( \frac{- E}{\kappa T} \right) \right] dE$$

(A6).

$G(E_0, \sigma)$ is the Gaussian distribution of activation energies with mean $E_0$ and standard deviation $\sigma$; $k_0$ is a frequency factor. Substituting (A1) for $t$ and 0.5 for $\Phi(t, T)$ gives:


$$\int_0^\infty G \left( E_0 , \sigma \right) \exp \left[ \frac{\kappa T_c{}^2 k_0}{E_A \left( dT / dt \right)} \exp \left( \frac{- E}{\kappa T_c} \right) \right] dE = 0.5$$

(A7).

(A7) is solved iteratively for $T_c$, given the model parameters $E_0$, $\sigma$ and $k_0$ in Table 1, and assuming a cooling rate $dT/dt$.



**Data Availability**

The Raman measurement data are available in the repository for digital research data of the TU Dresden and the TU Bergakademie Freiberg OPARA, at http://dx.doi.org/10.25532/OPARA-103.

**Author contribution**

BH and RJ planned the annealing experiments that were carried out by BH and BW. BH did the Raman measurements and the kinetic modeling. RJ and LR contributed to the interpretation of the models. BH prepared the manuscript with contributions from all co-authors.

**Competing interests**

The authors declare that they have no conflict of interests.

**Acknowledgments and Funding**

We thank Axel Schmitt (Universität Heidelberg) for providing the synthetic zircon. BH and BW are supported by a scholarship from the German Academic Scholarship Foundation (*Studienstiftung des deutschen Volkes*) and grant WA 4390/1-1 from the German Research Council (*Deutsche Forschungsgemeinschaft*), respectively.

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
