# Peer review of "The closure temperature(s) of zircon Raman dating"

_Geochronology, 2020_

## Referee Comment (RC1) · Beatrix Heller (Referee) · 4 Feb 2021

General remarks: The manuscript is written in a very scarce style omitting in some cases explications that would be necessary and details that would be interesting for the reader. This is especially true for the sample and measurement descriptions. The language is sometimes a bit imprecise, specific examples that need improvement are given in the provided supplement with specific remarks.

Taking into consideration that this manuscript is about a method which is not really established yet and only few examples exist on its application, the authors should write a bit more about the potential, possible applications and advantages of zircon Raman dating. Some citations would merit being included in the introduction. The idea of Ra-

diation damage dating goes actually back to the 1950s (see Holland and Kulp (1950): "Geologic Age from Metamict Minerals", Science, 111, p.312). Some important works of radiation damage might also be mentioned such as Holland and Gottfried (1955) and Murakami et al. (1991).

The authors seem to have a certain tendency to put trendlines into their data which, in some cases, seem to indicate rather what the authors want the reader so see than what the data actually shows. (For specific examples see the provided supplement)

The manuscript is about radiation damage and its annealing but the authors give not a single value of radiation damage density for their samples. In order to make the presented data comparable to other data, ideally U and Th concentrations should be measured for the analyzed spots and radiation damage densities should be calculated. If this is technically complicated the authors should at least estimate the damage densities from of their Raman spectra (e.g. by using the calibration by Palenik et al. 2003).

Calculation of the closure temperatures: in order to apply the obtained results to a lager set of samples it would be good it the authors could give additional values for the closure temperatures for very slow and very fast cooling (1°C/Ma and 100°C/Ma), 30°C/Ma seem less important though.

I did not check systematically but at least one citation (Palenik et al. 2003) is missing in the reference list. Please recheck

Several figures need some improvement and/or better explanations in the captions. For the concerned figure this is explained in detail in the provided supplement.

Please also note the supplement to this comment:
https://gchron.copernicus.org/preprints/gchron-2020-39/gchron-2020-39-RC1-supplement.pdf

**Supplement:**

Review of the manuscript entitled „The closure temperature(s) of zircon Raman dating" by Birk Härtel et al.
Referee: Beatrix Heller

Specific comments on the text and the figures:

L20: include also Palenik et al. 2003
L23: please define "high enough temperatures" and reformulate the sentence

L32: add Tc estimates "of the zircon Raman/damage thermochronometer…"

Figure 1: It would be nice to know the radiation damage density of the unannealed grain. If you want to be nice to colorblind readers do not use red and green in the same figure.

L39: What is your bandwidth? FWHM or HWHM? Please precise. Personally would prefer FWHM (rather than $\Gamma$) throughout the text as this is much more common practice in the zircon-damage-literature and would help the unfamiliar reader to better "read" the figures at one glance. But I understand that $\Gamma$ is shorter.

L44-46: Try to explain better, if I remember it right from the literature there are different types of point defects needing different energies to be annealed. From the current formulation I don't understand why some point defects anneal in stage I and others in stage III

L48: If I interpreted it right in the Data supplement you also measured other external bands. Either adapt the text ("..measured but not explored in detail but can be found in the supplement") or kick it out of the appendix

Figure 2: Please indicate where this accumulation trend comes from (we don't see the data it's based on) and what it means. Make maybe a nicer arrow, the arrowhead can be confused with a data point.

L58 and following: please specify "moderate to high" radiation damage densities, give numbers if possible. Age? U and Th contents? You should at least estimate the damage from the Raman spectrum. Compared to other samples I would not say that your samples are very metamict.

L59 and following: Try to better describe the grains: color? Size? Uniform sample? Zoning?

Section 2.2 (L63 and following): Give more information on the Raman measurements please. Acquisition times? Objective? Repeated measurements on the same spot, if yes how many, did you make any averaging? Did you measure several spots per grains? What is the excitated volume?
I don't get the need to cut the spectra into three to do a background correction. The same 3 sections you glued together before?

Section 2.3 (L70 and following): Please indicate how many grains you used for the experiments and how they were selected

Figure 3: The figure needs some improvement, please add a, b, c, d. I have a problem with the black trendlines (?) which seem to be added in a somehow aleatoric way. If you calculated a real trend, please say so. If it's just a rough indication please mention it, too, (the caption doesn't say anything) and maybe do not choose a black solid line but rather maybe sth gray, dashed and large? Especially for the v2 Band I would say that the trendline does not really fit the data. The picture below shows just as example an alternative option for a trend which fits the data equally good I would say. A more honest representation might be to draw very thin lines connecting the different annealing states of every grains. You could also include data points of synthetic zircon in the graphs. Like this it would make slightly more sense that the arrowheads go beyond the actual data. You could also choose a different shape for the unannealed samples

[Figure]

L90: changes in what?
L90: I don't see any break in slope between 700 and 800°C in Fig.3! I would rather say that the data overlaps pretty well… Between 600 and 800° eventually…

Figure 4: please add a, b, c, d. You should indicate the durations of the experiments in the captions or, if you find an elegant solution, directly in the graph. If I got it right the two experiments do not cover the same time range as there are only 5 blue but 7 red datapoints. Be honest with the reader and mention this somewhere if it's the case. And where do the knickpoints in the blue "trendlines" come from (esp. v2 and ER)??? There doesn't seem to be any data for this behavior. Assuming that at 1000°C the sample makes the same trend as at 600°C seems a bit too courageous to me. Again you could consider including the values for synthetic zircon.

Figure 5: It would be better to use the same colors as in Fig 3 and 8. Maybe extend the figure showing the same information for T vs bandwidth? I didn't test it but I can imagine that that could be interesting. In the end, this is what your model is based on.

L110:…"for which Geisler (2002) reported a constant value…" Can we see this somewhere? I don't. Or is this just the wrong citation and should be Zhang (2000)? Maybe indicate stages in Figure?

L120 and Figure 6: "…the unit cell shrinks anisotropically..": I have difficulties to see this in Figure 6. For the Geisler data maybe by omitting the two very scattered data points but for Colombo and Chrosh the data seems to be perfectly parallel to the lines with constant c/a ratio (especially if you consider a small error in the data which, unfortunately, is not presented). I am therefore not convinced if Figure 6 should be kept at all as it doesn't

present a very strong message. If you keep it please change the colors (purple and red are too similar) and you could gain some space and reduce the size by putting the legend in the lower right corner.

Figure 7: It could be interesting to indicate to which damage dose correspond the 12cm^-1 width

L148: rather calculated than fitted

L154: you should rather compare the different initial damage doses than the absolute age. Note that the latter, on a geological time scale is not so different as your ages are lower carboniferous and the samples of Ginster 2019 have U-Pb ages of max. 570Ma and He ages down to 414Ma.

Table 1: typo in pos [4,4]

Figure 10: b) Please explain in the legend or the caption what the filling colors in b) between the lines mean /show. Maybe choose different colors for that it becomes clearer.

Data Supplement: please, mention the existence of the supplement also in the text. Otherwise I think many readers might miss its existence.
Please fix the caption of Supplementary Table 2: $\phi$ explanation is missing
Also T2: There are some intermediate steps missing (e.g. sample 6 t=30 and t=1400). Why?
For the t=0 min steps you might replace the temperature by "unannealed" as in T1.

---

## Referee Comment (RC2) · Airton Dias (Referee) · 15 Mar 2021

Authors: Birk Hartel, Raymond Jonckheere, Bastian Wauschkuhn, and Lothar Ratschbacher Ms. Title: THE CLOSURE TEMPERATURE(S) OF ZIRCON RAMAN DATING

GENERAL COMMENTS:

I suggest that the article be ACCEPTED.

I have no serious corrections to the manuscript. It is very well written and the description of the results (content and figures) is excellent.

I make a single suggestion in order to increase the reference base of the manuscript

and a single question on a methodological level. In addition, I make some technical suggestions for improving the text.

SPECIFIC COMMENTS:

There are some points that should be addressed by the authors to improve the quality of the paper.

1) I suggest reading the article by Dias et al., 2020 (doi:10.1166/jnn.2020.17172). It is related to the content of this article. It may be an updated reference.

2) In the METHODS AND MATERIALS (2.2 RAMAN SPECTROMETRY), the laser used in the experiments is presented: 488 nm - line 64. I would like to know why this laser was used instead of laser regularly applied (514 and 633 nm)? What are the advantages of using laser 488 nm? Finally, I would like to receive more information to justify this choice. I even think that such information should be included in the text (even succinctly).

TECHNICAL CORRECTIONS:

• All manuscript: change "fission track" by "fission-track"

• Introduction, line 21: remove the word "of". It is unnecessary.

• Introduction, line 47: change "Our aim is" by "We aim".

• Annealing experiments, line 78: change "one hour" by "one-hour".

• Changes in band position and width, line 84: change "is" by "are".

• Changes in band position and width, line 89: insert "a" before "slope".

• Changes in band position and width, line 91: remove the word "the" before "stage". It is unnecessary.

• Changes in band position and width, line 97: remove the word "the" before "stage". It is unnecessary.

âĞ́ Changes in band position and width, line 104: insert "s" after "stage".

âĞ́ Changes in band position and width, line 123: remove the word "of" before "the decrease". It is unnecessary.

âĞ́ Kinetic modeling and closure temperature, line 154: change "age" by "ages".

âĞ́ Kinetic modeling and closure temperature, line 161: remove the word "an" before "activation". It is unnecessary.

âĞ́ Kinetic modeling and closure temperature, line 173: change "are" by "is".

âĞ́ Kinetic modeling and closure temperature, line 191: remove the word "the" before "faster". It is unnecessary.

âĞ́ Kinetic modeling and closure temperature, line 196: remove "," after "timescales".

âĞ́ Kinetic modeling and closure temperature, line 206: remove "a" before "lower". It is unnecessary.

âĞ́ Conclusions, line 249: change "for" by "of".

âĞ́ Conclusions, line 252: change "for" by "in".

---

## Author Comment (AC1) · 24 Mar 2021

MS: gchron-2020-39

**The closure temperature(s) of zircon Raman dating**

Härtel, B., Jonckheere, R., Wauschkuhn, B., and Ratschbacher, L.

**Replies to reviewer #1 (B. Heller)**

We thank the reviewer for her constructive comments. We are pleased that she did not find errros with our data or interpretations. We accepted her suggestions for a better presentation, in particular for a more detailed introduction and description of the samples and measurements. We address her specific comments (in italic) below:

| Comment | Reply |
|---|---|
| *Taking into consideration that this manuscript is about a method which is not really established yet and only few examples exist on its application, the authors should write a bit more about the potential, possible applications and advantages of zircon Raman dating. Some citations would merit being included in the introduction. The idea of Radiation damage dating goes actually back to the 1950s (see Holland and Kulp (1950): "Geologic Age from Metamict Minerals", Science, 111, p.312). Some important works of radiation damage might also be mentioned such as Holland and Gottfried (1955) and Murakami et al. (1991).* | We agree that Raman dating should be explained in more detail, and we included a paragraph describing the concept. |
| *The manuscript is about radiation damage and its annealing but the authors give not a single value of radiation damage density for their samples. In order to make the presented data comparable to other data, ideally U and Th concentrations should be measured for the analyzed spots and radiation damage densities should be calculated. If this is technically complicated the authors should at least estimate* | We included the initial damage densities calculated with the calibration of Váczi and Nasdala (2017). |

| | |
|---|---|
| *the damage densities from of their Raman spectra (e.g. by using the calibration by Palenik et al. 2003).* | |
| *Calculation of the closure temperatures: in order to apply the obtained results to a lager set of samples it would be good it the authors could give additional values for the closure temperatures for very slow and very fast cooling (1C/Ma and 100C/Ma), 30C/Ma seem less important though.* | We added additional closure temperatures for cooling rates of 1 °C/Myr and 100 °C/Myr to Table 1. |
| *I did not check systematically but at least one citation (Palenik et al. 2003) is missing in the reference list. Please recheck* | We checked the reference list and added missing references. |
| *L20: include also Palenik et al. 2003* | We included the references of the Raman-α-damage calibrations of Nasdala et al. (2001), Palenik et al. (2003), and Váczi and Nasdala (2017). |
| *L23: please define "high enough temperatures" and reformulate the sentence* | The term "high enough temperature" refers to the dependence of annealing temperature on annealing duration. The assumed temperature range for geological timescales is given in the following paragraph. We changed the sentence to "Radiation damage is annealed at elevated temperatures, with the exact temperature range depending on the annealing duration (Zhang et al., 2000a; Geisler et al., 2001; Nasdala et al., 2001; Pidgeon et al., 2016)." |
| *L32: add Tc estimates "of the zircon Raman/ damage thermochronometer..."* | We rephrased this sentence as "$T_c$ estimates for α-damage annealing..." |
| *Figure 1: It would be nice to know the radiation damage density of the unannealed grain. If you want to be nice to colorblind readers do not use red and green in the same figure.* | We added a radiation damage density to the caption and revised the colours used in our figures. |
| *L39: What is your bandwidth? FWHM or HWHM? Please precise. Personally would prefer FWHM (rather than Γ) throughout the text as this is much more common practice in the zircon-damage-literature and would help the unfamiliar reader to better "read" the figures at one glance. But I understand that Γ is shorter.* | We specified that we use FWHM at first appearance of the bandwidth in the revised manuscript. We adopted Γ for the bandwidth from Geisler et al. (2001) and Geisler (2002). The use of Γ is also consistent with the use of ω for peak position for which there is to our knowledge no convenient alternative. |
| *L44-46: Try to explain better, if I remember it right from the literature there are different types of point defects needing different energies to be annealed. From the current formulation I don't understand why some point defects anneal in stage I and others in stage III.* | It is indeed assumed that the activation energies are different for point defects associated with extension in the *a* and the *c* direction (Geisler, 2002). Ríos et al. (2000) proposed an annealing mechanism by point defect migration in the basal plane by tilting and twisting of the $ZrO_4$ polyhedra. The point defects responsible for extension in the *c* direction are considered more stable due to the stronger linkage of the $SiO_4$ and $ZrO_4$ polyhedra. Geisler (2002) associated the elimination of these defects during re-alignment of the tilted polyhedra with the high activation energies for stage III annealing. We included this explanation in the manuscript. |

| | |
|---|---|
| *L48: If I interpreted it right in the Data supplement you also measured other external bands. Either adapt the text (".measured but not explored in detail but can be found in the supplement") or kick it out of the appendix* | We added a reference to these data in the text. |
| *Figure 2: Please indicate where this accumulation trend comes from (we don't see the data it's based on) and what it means. Make maybe a nicer arrow, the arrowhead can be confused with a data point.* | The damage accumulation trend is based on measurements of unannealed grains. From these, we selected the grains for the annealing experiments (Figures 3 and 4). We included them in the Figure. |
| *L58 and following: please specify "moderate to high" radiation damage densities, give numbers if possible. Age? U and Th contents? You should at least estimate the damage from the Raman spectrum. Compared to other samples I would not say that your samples are very metamict.* | The zircons were separated from the Oederan Forest subtype of the Schweddey ignimbrite in the Saxonian Flöha Basin. This is equivalent to the samples MfNC-2014-01 and MfNC-2014-02, for which Löcse et al. (2019) report zircon U-Pb ages of 309.0 ± 1.8 Ma and 309.4 ± 2.6 Ma. The most damaged zircon in this study has a $\nu_3(SiO_4)$ bandwidth $\Gamma_3 \approx 24$ cm$^{-1}$, corresponding to $D_\alpha \approx 200 \cdot 10^{16}$ α/g using the calibration of Váczi and Nasdala (2017), and roughly to the first percolation point of Salje et al. (1999), as recalculated by Nasdala et al. (2004). We classify this as representing a high damage density. The least damaged zircon of the sample can be classified as slightly damaged zircon with $\Gamma_3 \approx 5.5$ cm$^{-1}$ ($D_\alpha \approx 22 \cdot 10^{16}$ α/g). We added this information to section 2.1. |
| *L59 and following: Try to better describe the grains: color? Size? Uniform sample? Zoning?* | The zircons are prismatic in shape with dominant {100} prisms and {101} pyramids. All grains are transparent or translucent, colourless to brownish-red. Some have inclusions of apatite or quartz. The long axis lengths range from ~150 to ~300 µm. Löcse et al. (2019) described oscillatory and sector-zoned zircons from this ignimbrite. We included the above information in section 2.1. |
| *Section 2.2 (L63 and following): Give more information on the Raman measurements please. Acquisition times? Objective? Repeated measurements on the same spot, if yes how many, did you make any averaging? Did you measure several spots per grains? What is the excitated volume?* | We carried out a single measurement per spot, acquiring 10 spectra for 20 s each, so that for the three bandwidth intervals recorded for each full spectrum, we had a total acquisition time of 10 minutes. We used an Olympus BX 51WI microscope with an Olympus 50x (n.a. 0.75) objective. We measured one spot in most grains, but in some grains we measured one spot at the center and one at the rim. The spot size on the sample was ~ 2 µm; the depth likely exceeds 2 µm. Grains that showed additional Raman bands from inclusions or asymmetric bands due to overlap of Raman signals from low- and high-damage zones (Nasdala et al., 2005) were excluded. We included this additional information in section 2.2. |
| *I don't get the need to cut the spectra into three to do a background correction. The same 3 sections you glued together before?* | The sections glued and the sections fitted are not the same. The sections glued together in the Step-and-Glue measurement are determined by the position and width of the detector. |

| | For background correction and peak fitting, we divided the spectrum into three groups of Raman bands so that the background fit for one part does not influence the background correction in other parts. We included the explanation of the Step-and-Glue algorithm in the manuscript. |
|---|---|
| *Section 2.3 (L70 and following): Please indicate how many grains you used for the experiments and how they were selected* | We selected 6-12 grains for each annealing run to cover a broad range of initial α-damage densities, estimated from the pre-annealing measurements. |
| *Figure 3: The figure needs some improvement, please add a, b, c, d.* | Done. |
| *I have a problem with the black trendlines (?) which seem to be added in a somehow aleatoric way. If you calculated a real trend, please say so. If it's just a rough indication please mention it, too, (the caption doesn't say anything) and maybe do not choose a black solid line but rather maybe sth gray, dashed and large? Especially for the v2 Band I would say that the trendline does not really fit the data. The picture below shows just as example an alternative option for a trend which fits the data equally good I would say. A more honest representation might be to draw very thin lines connecting the different annealing states of every grains.* | We included a shaded arrow as optical guidance. The boundaries of the shaded area are parallel to the trends of the most and the least damaged zircons during annealing (see below) to encompass the bundle of single annealing paths. We also included the synthetic zircon. |
| *You could also include data points of synthetic zircon in the graphs. Like this it would make slightly more sense that the arrowheads go beyond the actual data. You could also choose a different shape for the unannealed samples* | We also included the synthetic zircon and made the symbols of the unannealed starting data of the annealing runs transparent. |

[Figure]

| *L90: changes in what?* | We changed the sentence to "We observed the greatest changes in $\omega_3$ and $\Gamma_{ER}$." |

| | |
|---|---|
| *L90: I don't see any break in slope between 700 and 800°C in Fig.3! I would rather say that the data overlaps pretty well… Between 600 and 800° eventually…* | The samples at 700 °C are consistent with the trend from the unannealed samples to the 600 °C runs, but also with the flat trend of the samples annealed at higher temperatures. We changed the sentence to "We interpret the change in the slope at ~ 700 °C as the transition from stage I to stage II annealing (Geisler et al., 2001)." |
| *Figure 4: please add a, b, c, d.* | Done. |
| *You should indicate the durations of the experiments in the captions or, if you find an elegant solution, directly in the graph. If I got it right the two experiments do not cover the same time range as there are only 5 blue but 7 red datapoints. Be honest with the reader and mention this somewhere if it's the case.* | We included the duration of the experiments in the caption. We also added the difference in time-steps, since two of the measurements at 1000 °C was discarded as the zircon broke during the 5-day experiment. |
| *And where do the knickpoints in the blue "trendlines" come from (esp. v2 and ER)??? There doesn't seem to be any data for this behavior. Assuming that at 1000°C the sample makes the same trend as at 600°C seems a bit too courageous to me.* | The knickpoints for the cyan lines are outside the data range because at 1000 °C, the zircons already reach stage II at short annealing durations. We assumed the trend for stage I to be approximately parallel to the data of the 600 °C experiment, based on Geisler et al. (2001), Geisler (2002), and Ginster (2019) whose trends in stage I did not vary for a range of temperatures.

 We changed the style of the first segment of the cyan trajectory to a dashed line to indicate that it is not based on the annealing data. |
| *Again you could consider including the values for synthetic zircon.* | Done. |

[Figure]

| | |
|---|---|
| *Figure 5: It would be better to use the same colors as in Fig 3 and 8.* | We changed the colours to match the temperatures in Fig. 3, 4, 5 and 8. |
| *Maybe extend the figure showing the same information for T vs bandwidth? I didn't test it* | We do not show a comparison of bandwidth vs. T, because Zhang et al. (2000) did not |

| | |
|---|---|
| *but I can imagine that that could be interesting. In the end, this is what your model is based on.* | provide bandwidths for their samples. Furthermore, the transition from one annealing stage to the next is more conspicuous in $\omega$ than in $\Gamma$. |
| *L110:…"for which Geisler (2002) reported a constant value…" Can we see this somewhere? I don't. Or is this just the wrong citation and should be Zhang (2000)? Maybe indicate stages in Figure?* | We rephrased the citation to: "for which Geisler (2002) reported a sharp increase only in stage III. The data of Zhang et al. (2000a) show a slight decrease in the first two stages that reverses in stage III." |
| *Maybe indicate stages in Figure?* | We added labels for the stages in the revised version of the figure. |
| *L120 and Figure 6: "…the unit cell shrinks anisotropically..": I have difficulties to see this in Figure 6. For the Geisler data maybe by omitting the two very scattered data points but for Colombo and Chrosh the data seems to be perfectly parallel to the lines with constant $c/a$ ratio (especially if you consider a small error in the data which, unfortunately, is not presented).* | We agree that, due to the scatter, the trends do not show a definite increase in $c/a$. Still, about half of the data of Geisler (2002) show $c/a$ ratios above those for the initial, damaged zircon and the data of Colombo and Chrosch (1998a) show a convex curvature instead of a linear trend parallel to the $c/a$ isolines. In our opinion, the unit cell data of Colombo & Chrosch (1998a) and Geisler (2002) do not contradict our interpretation but we admit that they do not prove it either. |
| *I am therefore not convinced if Figure 6 should be kept at all as it doesn't present a very strong message. If you keep it please change the colors (purple and red are too similar) and you could gain some space and reduce the size by putting the legend in the lower right corner.* | We agree that our hypothesis concerning the downshift of $\omega_2$ is not the central message of this manuscript, although not without interest. We moved the figure and discussion to an appendix, and change the colours. |
| *Figure 7: It could be interesting to indicate to which damage dose correspond the 12cm^-1 width* | We added the damage density ($\sim 70 \cdot 10^{16}$ $\alpha/$g) to the figure caption. |
| *L148: rather calculated than fitted* | Done. |
| *L154: you should rather compare the different initial damage doses than the absolute age. Note that the latter, on a geological time scale is not so different as your ages are lower carboniferous and the samples of Ginster 2019 have U-Pb ages of max. 570Ma and He ages down to 414Ma.* | We agree that the damage accumulation times of our Flöha Basin zircons and the Sri Lanka zircons of Ginster et al. (2019) are not too different. It is however known that the Sri Lanka zircons have been partially annealed during their geological history, although the timing and the processes involved are unclear. In contrast, the zircons from the Flöha Basin are assumed to be unannealed, similar to the Chemnitz Basin zircons of Nasdala et al. (1998). We emphasize that Geisler et al. (2001), Geisler (2002), and Ginster et al. (2019) used zircons from Sri Lanka. Our data from geologically unannealed zircons set an independent constraint on $\alpha$-damage annealing measured with Raman. We included the above information in our discussion. |
| *Table 1: typo in pos [4,4]* | Done. |
| *Figure 10: b) Please explain in the legend or the caption what the filling colors in b) between the lines mean /show. Maybe choose different colors for that it becomes clearer.* | The filling colours mark the temperature intervals of the three different PAZ. We added this information to the caption. |
| *Data Supplement: please, mention the exist-* | We included references to the Supplement in |

| | |
|---|---|
| *ence of the supplement also in the text. Otherwise I think many readers might miss its existence.* | the Results & Discussion Section. |
| *Please fix the caption of Supplementary Table 2: Φ explanation is missing* | We added this description to the Supplementary Table. |
| *Also T2: There are some intermediate steps missing (e.g. sample 6 t=30 and t=1400). Why?* | We rejected measurements, when we had reason to doubt that they were in the same spot as the earlier measurements or when the spectra showed asymmetric bands. We included this information in the Methods section. |
| *For the t=0 min steps you might replace the temperature by "unannealed" as in T1.* | We changed the respective cells to "-". |

---

## Author Comment (AC2) · 24 Mar 2021

**TECHNISCHE UNIVERSITÄT BERGAKADEMIE FREIBERG**

MS: gchron-2020-39

**The closure temperature(s) of zircon Raman dating**

Härtel, B., Jonckheere, R., Wauschkuhn, B., and Ratschbacher, L.

**Replies to referee #2 (A. Dias)**

We thank the reviewer for his suggestions. We are pleased that the reviewer accepts our data and interpretation. We implemented his suggestions to extend our references and add technical details of the measurement setup. We address the specific comments (in italic) below:

| Comment | Reply |
|---|---|
| *I suggest reading the article by Dias et al., 2020 (doi:10.1166/jnn.2020.17172). It is related to the content of this article. It may be an updated reference.* | We read the article and integrated relevant information in our Results & Discussion Section. |
| *In the METHODS AND MATERIALS (2.2 RAMAN SPECTROMETRY), the laser used in the experiments is presented: 488 nm - line 64. I would like to know why this laser was used instead of laser regularly applied (514 and 633 nm)? What are the advantages of using laser 488 nm? Finally, I would like to receive more information to justify this choice. I even think that such information should be included in the text (even succinctly).* | We used the 488 nm laser to increase the signal-to-noise ratio of our spectra. Since the Raman intensity is proportional to $1/\lambda^4$ of the incident laser light, the 488 nm laser gives higher Raman intensities than a 514 or 633 nm laser, although at lower spectral resolution. Using this laser lowered the acquisition times of our measurements. |
| *All manuscript: change "fission track" by "fission-track"* | Done. |
| *Introduction, line 21: remove the word "of". It is unnecessary.* | Done. |
| *Introduction, line 47: change "Our aim is" by "We aim".* | Done. |
| *Annealing experiments, line 78: change "one hour" by "one-hour".* | Done. |

| | |
|---|---|
| *Changes in band position and width, line 84: change "is" by "are".* | Done. |
| *Changes in band position and width, line 89: insert "a" before "slope".* | Done. |
| *Changes in band position and width, line 91: remove the word "the" before "stage". It is unnecessary.* | Done. |
| *Changes in band position and width, line 97: remove the word "the" before "stage".*
*It is unnecessary.* | Done. |
| *Changes in band position and width, line 104: insert "s" after "stage".* | Done. |
| *Changes in band position and width, line 123: remove the word "of" before "the decrease". It is unnecessary.* | Done. |